# AN-Net: An Anti-Noise Network For Anonymous Traffic Classification

## ABSTRACT

Anonymous networks employ a triple proxy to transmit packets to enhance user privacy, causing traffic packets from all applications and web services to form a unified flow. The traditional approach of applying flow-level encrypted traffic classification methods to anonymous traffic (i.e., treating consecutive packets as a single flow) is hindered by irrelevant packet noise. Moreover, fluctuations in the network environment can introduce per-packet attribute noise and discrepancies between training and test data. How to extract robust patterns from consecutive packets replete with noise remains a key challenge. In this paper, we propose the **A**nti-**N**oise **N**etwork (**AN-Net**) to construct robust short-term representations for a single modality, effectively countering irrelevant packet noise. We also incorporate an enhanced multi-modal fusion approach to combat per-packet attribute noise. AN-Net achieves state-of-the-art performance across two anonymous traffic classification tasks and one VPN traffic classification task, notably elevating the F1 score of SJTU-AN21 to 94.39% (6.24%↑). In particular, attackers cannot easily disrupt all short-term features of all modalities and thus AN-Net is robust against injected noise packet attacks. Our codes will be available on GitHub after the double-blind review process.

## CCS CONCEPTS

• **Information systems** → **Traffic analysis**; • **Security and privacy** → *Network security*; • **Computing methodologies** → *Artificial intelligence*.

## KEYWORDS

Anonymous Traffic Classification, Irrelevant Packet Noise, Per-Packet Attribute Noise, Short-Term Representation, Multi-Modal Fusion

### ACM Reference Format:
Anonymous Author(s). 2018. AN-Net: An Anti-Noise Network For Anonymous Traffic Classification. In *Proceedings of Make sure to enter the correct conference title from your rights confirmation emai (Conference acronym 'XX).* ACM, New York, NY, USA, 10 pages. https://doi.org/XXXXXXX.XXXXXXX

## 1 INTRODUCTION

Network traffic classification, aiming at classifying network traffic from various applications or web services, plays a critical role in quality of service (QoS) enhancement, resource usage planning,

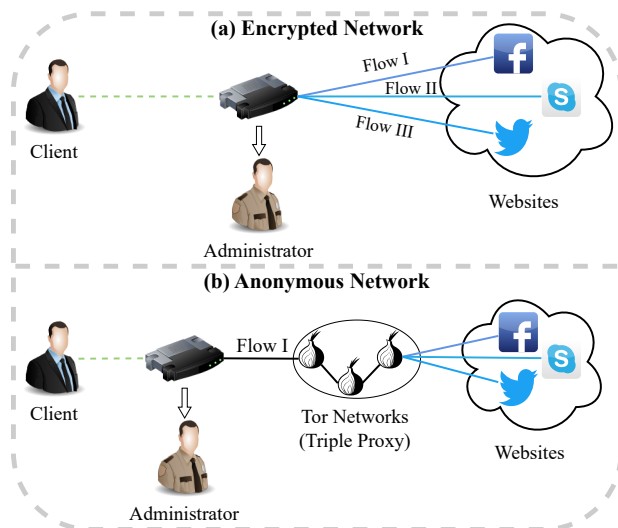

**Figure 1: Threat models of encrypted network traffic classification and anonymous network traffic classification.**

and even malware detection [6, 31, 35]. Recently, various traffic encryption techniques have been employed [18, 28], such as SSL, for protecting user privacy. However, these encryption mechanisms also help malicious traffic evade the surveillance system, thus bringing great challenges to traffic classification [7]. Traditional deep packet inspection (DIP) based methods [15, 32], which explore regular expression for matching the payload data, fail to identify encrypted traffic since the payload data is changed relying on the encryption algorithm. Therefore, encrypted traffic classification has become a research hotspot in recent years.

Over the past decade, many different methods have been proposed to classify encrypted traffic, which can be divided into three categories according to the types of input: statistical feature-based methods, sequential attribute-based methods, and raw traffic-based methods. Early works [2, 3, 12, 13, 37] extract the statistical features at the flow level (e.g., mean, minimum, maximum, and standard deviation of packet sizes in a flow) to train the machine learning-based classifier. These methods rely on expert-designed features heavily and have limited generalization ability. Recently, some deep learning-based methods [24, 26, 34] automatically learn complicated patterns from the raw flow attribute sequences (e.g., packet sizes in a flow), and achieve significant performance improvement. However, these methods require a large number of labeled data to train the deep learning-based models for more robustness. As a comparison, raw traffic-based methods [25, 42] directly capture the implicit and robust patterns in the encrypted payload at flow level using complicated models. In addition to large amounts of labeled

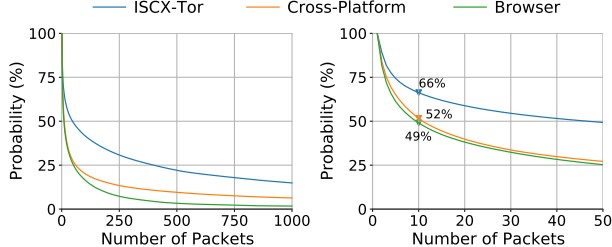

**Figure 2: The relationship between the number of consecutive packets and the probability that they belong to the same flow. Consecutive packets in short-term are likely to originate from the same flow.**

data, these methods are also limited by long training time and high requirements on computing resources.

Most importantly, the majority of encrypted traffic classification methods described above is based on flow-level features. Flow level aggregation is beneficial for extracting robust patterns, but also limits the effectiveness in special cases, e.g., the anonymous network. As illustrated in Figure 1, tor network, the most mainstream anonymous network, uses triple proxy to transmit traffic packets for protecting user privacy. As a result, traffic packets from all applications or web services form a single flow. The conventional approach of applying flow-level encrypted traffic classification methods to anonymous traffic is to take consecutive packets as a flow [45]. Obviously, this approach cannot guarantee that all packets originate from the same web service. Therefore, the key difference between anonymous traffic and encrypted traffic is noise, i.e., irrelevant packets from other flows, denoted as **irrelevant packet noise**.

Extracting robust patterns from consecutive packets full of noisy packets is crucial for anonymous traffic classification. Fu *et.al.* [17] found that most flows completed in less that 2 seconds, which indicates that consecutive packets in short-term are likely to originate from the same flow. Inspired by their observation, we visualize the number of consecutive packets and their probability of belonging to the same flow on ISCX-nonTor [22], Cross-Platform [38], and Browser datasets (see Figure 2). For simplicity, we use the first $1 \times 10^4$ packets of each pcap file to plot the figure. Results show that consecutive packets in long-term have a high probability of not belonging to the same flow, but packets in short-term are likely to originate from the same. Therefore, anonymous traffic classification methods should learn to model short-term features with low noise and then aggregate them for robust representation.

In addition to irrelevant packet noise, fluctuations in the network environment can introduce noise in the per-packet attributes, denoted as **per-packet attribute noise**. For example, internal arrival time (IAT) is likely to be affected by network congestion and time-to-live (TTL) may also change due to network routing. Therefore, extracting robust patterns from noisy per-packet attributes is also important for anonymous traffic classification. Considering that traffic packets usually have more than one attribute, e.g., packet size, IAT, TTL, etc., a good idea to combat the interference of per-packet attribute noise is to combine attributes from different modalities.

In this paper, we propose an **A**nti-**N**oise **Net**work (**AN-Net**) for classifying anonymous traffic via short-term representation building and enhanced multi-modal fusion. It aims to learning robust patterns from anonymous traffic full of irrelevant packet noise and per-packet attribute noise. We first propose a Uni-modal Short-term Representation Learning Module. It divides consecutive packets in a "flow" into multiple short-term packet sequences. Short-term features are then extracetd from them by using the Short-term Feature Extraction Module (SFEM). Once short-term features are extracted, they are fed into the Short-term Representation Aggregation Module (SRAM), which aggregates the short-term features into flow-level representation. The SRAM is specifically designed to identify which short-term features come from irrelevant flows and which ones originate from the target flow by adopting a novel high temperature self-attention mechanism, thus helps resist the irrelevant packet noise. Finally, flow-level representations from different modalities are fused in the Enhanced Multi-modal Representation Fusion Module to combat the per-packet attribute noise.

The main contributions of this paper are summarized as follows:

- We present a Uni-modal Short-term Representation Learning Module to construct robust short-term representations for a single modality to resist irrelevant packet noise. We design a novel high temperature self-attention mechanism, which pays less attention to noise packets.
- We propose to fuse representations from different modalities to combat per-packet attribute noise. A novel representation enhancement strategy is employed to further improve fusion performance.
- AN-Net achieves state-of-the-art performance over two anonymous traffic classification tasks and one VPN traffic classification task. Moreover, it exhibits strong robustness against injected noise packet attacks.

## 2 RELATED WORK

### 2.1 Conventional Traffic Classification

Port-based methods [30] identify the application type based on the port used. Their efficiency declined with the increase use of dynamic ports [8] and default ports [14]. Payload-based methods [6, 15, 21, 32, 33], also called deep packet inspection (DPI), explore the specific signature strings for matching the payload data. These methods are unable to classify encrypted traffic because the signature strings cannot be obtained from payloads after encryption.

### 2.2 Encrypted Traffic classification

Encrypted traffic classification methods can be divided into three categories according the the types of input: statistical feature-based methods, sequential attribute-based methods, and raw traffic-based methods.

*2.2.1 Machine Learning Based Encrypted Traffic Classification.* Most statistical feature-based methods use ML-based models for classification. These methods propose to leverage statistical features at flow-level (e.g., mean, minimum, maximum, and standard deviation of packet sizes in a flow) to solve encrypted traffic classification problem combined with machine learning algorithms [1, 4, 12, 27, 37]. AppScanner [37] trains random forest classifiers by

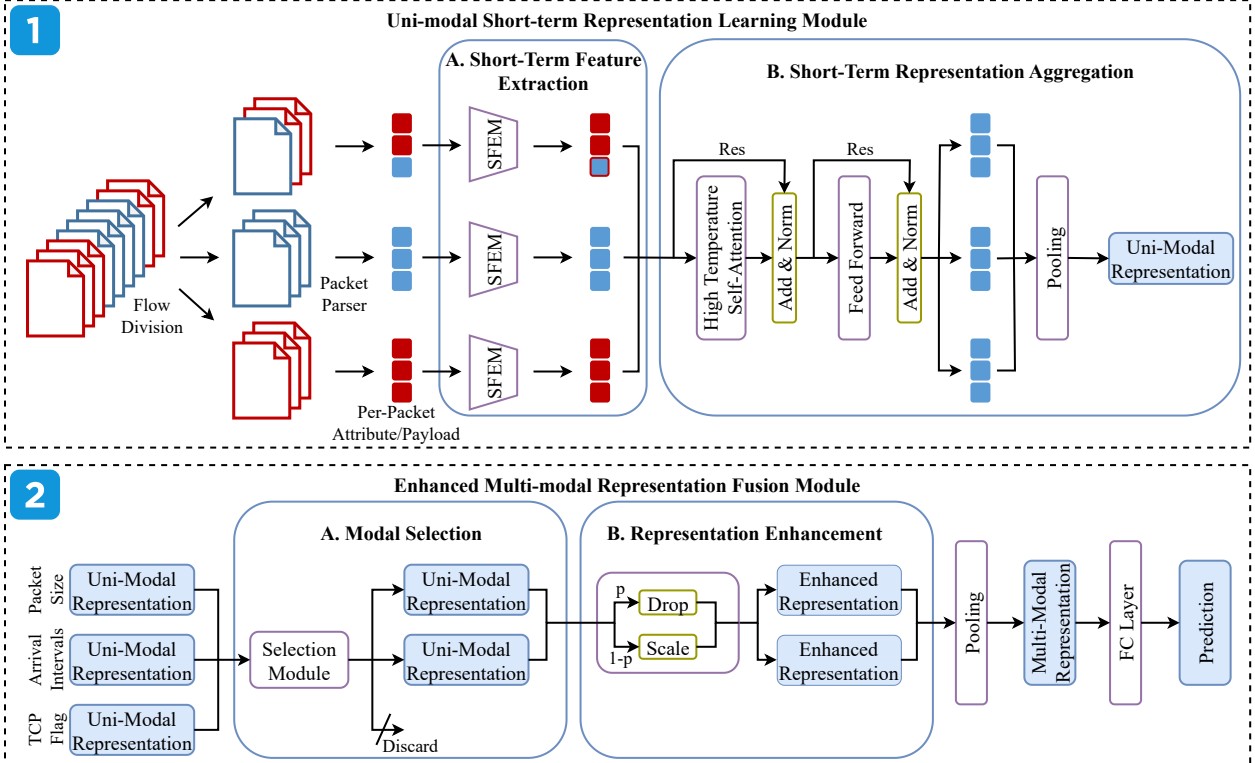

**Figure 3: Overview of AN-Net Framework.**

exploiting statistical features of packet sizes, while Gerard *et.al.* [12] trains C4.5 decision tree and KNN classifiers using time-related features. As a supplement of statistical features, Whisper [16] extracts the frequency domain features of flows and uses clustering algorithms for classification. These methods rely heavily on professional knowledge and it is difficult to design generic statistical features to handle different applications.

*2.2.2 Deep Learning Based Encrypted Traffic Classification.* Some statistical feature-based methods [44] also apply DL-based models for better representation extraction capabilities, and they also rely on human-designed features and have limited generalization ability. As an alternative, sequential attribute-based methods [5, 24, 26, 34, 36] extracts discriminative representations from raw sequential attributes (e.g., packet sizes in a flow). Flowlens [5] computes for each flow a memory-efficient representation of packet sizes named "flow marker". FlowPic [34] transforms raw packet size sequences and arrival interval sequences in a flow into an intuitive picture. FS-Net [26] uses recurrent neural networks (RNN) to automatically extract representations from raw packet size sequences. Another alternative approach is to learning implicit representations from raw traffic. Raw traffic-based methods [24, 25, 29, 42] directly capture the implicit and robust patterns in the encrypted payload at flow level using complicated DL-based models.

However, the majority of encrypted traffic classification methods described above is based on flow-level features, which limits their effectiveness in anonymous networks, where the traffic from all applications or web services form a single flow. Moreover, None

of these methods paid attention to the unreliability of per-packet attributes and attempted to solve it by combining information from different modalities. In this paper, we propose to build strong short-term representations to resist irrelevant packet noise, and adopt an enhanced multi-modal fusion module to combat per-packet attribute noise.

## 3 AN-NET

In this paper, we aim to accurately classify anonymous network traffic under the interference of irrelevant packet noise and per-packet attribute noise. To this end, we propose an Anti-Noise network (AN-Net) (see Figure 3) to build strong short-term representations for a single modality to resist irrelevant packet noise (Secion 3.1) and achieve enhanced multi-modal fusion to combat per-packet attribute noise (Secion 3.2).

### 3.1 Short-term Representation Learning

In this section, we propose a Uni-modal Short-term Representation Learning Module to build short-term representations for resisting irrelevant packet noise.

*3.1.1 Flow Division and Packet Parsing.* Given a "flow" that consists of consecutive packets $P$ of length $L$, we first divide $P$ into $N$ parts and obtain multiple short-term consecutive packet sequences: $P = [P_1, P_2, \cdots, P_N]$. As illustrated in Figure 2, packets in short-term are likely to originate from the same flow. Therefore, features extracted from short-term packet sequences are more likely to be

immune to the interference of irrelevant packet noise. Then we parse out the short-term per-packet attribute/payload sequences for classification, denoted as $A = [A_1, A_2, \cdots, A_N]$, where $A_i$ is a short-term per-packet attribute/payload sequence.

*3.1.2 Short-term Feature Extraction.* We design two Short-term Feature Extraction Modules (SFEM) to extract short-term features from raw data and statistical data respectively, denoted as Raw-SFEM and Stat-SFEM.

Suppose the input is a short-term per-packet payload sequence $A_i \in \mathbb{R}^{l \times d}$, where $l$ is the length of the short-term sequence and $d$ is the length of payload, Raw-Stat employs a bidirectional GRU to extract the short-term feature: $F_i = GRU(A_i) \in \mathbb{R}^C$. If the input is a short-term per-packet attribute sequence $A_i \in \mathbb{R}^{l \times 1}$ (e.g., packet size), Raw-Stat first embeds each attribute to a vector via an embedding layer, and then also uses a bidirectional GRU to extract the short-term feature $F_i \in \mathbb{R}^C$. The extracted short-term features are denoted as $F = [F_1, F_2, \cdots, F_N] \in \mathbb{R}^{N \times C}$.

Stat-SFEM can only deal with attribute sequences. It first extracts 7 general statistical features (i.e., mean, max, min, median, standard deviation, skewness, and kurtosis) and the frequency domain features from these short-term attribute sequences as the short-term statistical features, denoted as $T = [T_1, T_2, \cdots, T_N]$. Then it employs a MLP to extract the short-term feature $F_i$ for each short-term statistic feature: $F_i = MLP(T_i) \in \mathbb{R}^C$. The MLP consists of two fully-connected layers and one ReLU layer between them. Despite its simplicity, the MLP is able to extract discriminative features thanks to the high-level statistical features and the nonlinear transformation of the ReLU layer. Thanks to the high-level statistical features, Stat-SFEM exhibits higher stability than Raw-SFEM when training data collection environment is inconsistent with the actual test environment, as detailed in Section 4.5.

*3.1.3 Short-term Representation Aggregation.* Once short-term features $F \in \mathbb{R}^{N \times C}$ are extracted, we use the Short-term Representation Aggregation Module (SRAM) to aggregate the short-term features $F \in \mathbb{R}^{N \times C}$ into flow-level representation $Z$. Since short-term features may also come from irrelevant flows, it is critical to distinguish among $N$ short-term features which ones originate from the irrelevant flow and which ones come from the target flow. We design a novel high temperature self-attention mechanism to achieve this. Specifically, the SRAM is composed of a Transform Layer [39] and a Pooling layer. A normal Transformer layer consists of two key sub-layers: self-attention layer and feed forward layer. Each sub-layer uses the residual structure [19] to avoid the degradation problem that occurs as the depth of the network increases.

Given the short-term features $F = [F_1, F_2, \cdots, F_N] \in \mathbb{R}^{N \times C}$, the self-attention layer first calculate Query Matrix $Q$ by using linear transformation:

$$Q = FW^Q = [q_1, q_2, \cdots q_N]^T, \quad (1)$$

where $W^Q \in \mathbb{R}^{C \times D}$ is learnable parameter and $q_i \in \mathbb{R}^D$ denotes the query vector of i-th short-term feature $F_i$. Similarly, we calculate Key Matrix and Value Matrix by another two linear transformations:

$$K = FW^K = [k_1, k_2, \cdots k_N]^T, \quad (2)$$

$$V = FW^V = [v_1, v_2, \cdots v_N]^T, \quad (3)$$

where $k_i$ denotes the key vector and $v_i$ denotes the value vector of i-th short-term feature $F_i$. Then, considering the query vector of i-th short-term feature $q_i$, we compute the dot products of $q_i$ and key vectors of all short-term features:

$$S_i = [q_i \cdot k_1^T, q_i \cdot k_2^T, \cdots, q_i \cdot k_N^T] = [s_{i1}, s_{i2}, \cdots, s_{iN}], \quad (4)$$

where $s_{ij}$ is the similarity between $q_i$ and $k_j$, and reflects the importance of j-th short-term feature to i-th short-term feature. Normal self-attention layer scales dot products $S_i$ by $\frac{1}{\sqrt{D}}$ before applying a softmax function to make the sum of the elements be 1:

$$\mathcal{W}_i = softmax(\frac{S_i}{\sqrt{D}}) = [w_{i1}, w_{i2}, \cdots, w_{iN}], \sum_{j=1}^{N} w_{ij} = 1, \quad (5)$$

The output at i-th position is then calculated using weighted summation over value vectors of all short-term features $V$:

$$z_i = \mathcal{W}_i V = \sum_{j=1}^{N} w_{ij} v_j. \quad (6)$$

Finally, the output of self-attention layer on all short-term features $F$ is represented as: $SelfAttn(F) = [z_1, z_2, \cdots, z_N]$. The above process can also be expressed in matrix form:

$$SelfAttn(F) = softmax(\frac{QK^T}{\sqrt{D}})V. \quad (7)$$

As stated above, the original self-attention layer scales the dot products $S_i$ by $\frac{1}{\sqrt{D}}$ before applying a softmax function. Wei *et.al.* [41] demonstrated that increasing the magnitude $||S_i||$ will cause a sharp distribution for softmax weight score $\mathcal{W}_i$. Original self-attention layer reduces the magnitude $||S_i||$ by scaling by $\frac{1}{\sqrt{D}}$ to avoid the softmax function from producing extremely small weights in $\mathcal{W}_i$. However, in the anonymous traffic classification scenario, since some short-term features may come from irrelevant flows, a sharp distribution for softmax weight score $\mathcal{W}_i$ needs to be generated to resist irrelevant packet noise. To this end, we design a novel high temperature self-attention mechanism by increasing the magnitude of dot products:

$$HT\text{-}SelfAttn(F) = softmax(\frac{\mathcal{N}(Q)\mathcal{N}(K)^T}{\tau})V, \quad (8)$$

where $\mathcal{N}$ is the normalize function that makes the vector norm equal to 1 and $\tau$ is the temperature hyper-parameter. Note that the real temperature is the reciprocal of $\tau$. After employing a high temperature, the softmax function produces extremely small weights for short-term features from irrelevant flows.

The feed forward layer can enhance the expression ability of the output features by mapping them to high-latitude space and then back to low-latitude space through two linear transformations. In the middle of them, a GeLU layer [20] is adopted to alleviate the vanishing gradient problem.

Finally, the short-term features are aggregated into flow-level uni-modal representation $Z$ by using an Average Pooling layer.

## 3.2 Multi-modal Representation Fusion

In this section, we propose the Enhanced Multi-modal Representation Fusion Module to fuse flow-level representations from different modalities for combating per-packet attribute noise.

*3.2.1 Modal Selection.* Before fusing representations from different modalities, we resort to information leakage [23] to remove useless modalities. We utilize the mutual information between statistical features $T$ of each modality and the ground truth labels $C$ to measure the importance of this modality:

$$I(T; C) = H(C) - H(C|T). \tag{9}$$

Representations from modalities with high information leakage are then fused to obtain the final robust representation.

*3.2.2 Representation Enhancement.* As mentioned above, the unreliability of per-packet attributes may make the representation of certain modalities full of noise. A conventional approach to combat input noise is to employ data augmentation strategy, which has been widely used in CV [9–11] and NLP [40, 43]. However, due to the limitation of input data type (i.e., numeric values), few data augmentation methods have been proposed to cope with encrypted traffic classification. To this end, we propose a novel representation enhancement strategy to perform data augmentation in representation-level.

Given representations from $M$ modalities $[Z_1, Z_2, \cdots, Z_M]$, we perform data augmentation on representation from each modality, respectively:

$$\hat{Z}_i = \begin{cases} 0, & p \\ B \times Z_i, & 1 - p \end{cases} \tag{10}$$

where $p$ is a random probability and $B$ is a scaling factor sampled from Beta distribution. Note that the representation-level data augmentation will only be adopted during the training phase. For a uni-modal representation, we randomly drop it to force the model to learn from other modalities, or scale it to make the model learn more robust patterns.

*3.2.3 Representation Fusion.* We then aggregate the enhanced representations from different modalities into the final multi-modal representation by using an Average Pooling layer:

$$\overline{Z} = AvgPooling([\hat{Z}_1, \hat{Z}_2, \cdots, \hat{Z}_M]). \tag{11}$$

Finally, we make the prediction on it with a fully-connected layer $\hat{Y} = FC(\overline{Z})$, and train the whole model through cross-entropy loss: $\mathcal{L} = CE(\hat{Y}, Y)$, where $Y$ is the ground-truth label.

# 4 EXPERIMENTS

In this section, we first present the datasets, baselines, evaluation metrics and implementation details (Section 4.1). We then compare AN-Net with seven methods (Section 4.2), and demonstrate that AN-Net is robust against injected noise packet attacks (Section 4.3). We further perform an ablation analysis of two key structures: Short-term Representation Learning (Section 4.4.1) and Multi-modal Representation Fusion (Section 4.4.2). Finally, we show that high-level statistical features are more stable when there are discrepancies between training and test data (Section 4.5).

## 4.1 Experiment Setup

*4.1.1 Datasets.* To evaluate the effectiveness and generalization of AN-Net, we conduct experiments on two anonymous traffic datasets [22, 44] and one VPN traffic dataset [12]. The statistical information of three datasets is shown in Table 1. Note that we take 100 consecutive packets as a flow. SJTU-AN21 provides a test set

**Table 1: The Statistical Information of three Datasets.**

| Dataset | Type | #Flows | | #Label |
| --- | --- | --- | --- | --- |
| | | train | test | |
| SJTU-AN21 [45] | Anonymous | 37529 | 9133 | 10 |
| ISCX-Tor [22] | Anonymous | 92458 | 23101 | 8 |
| ISCX-VPN [12] | VPN | 20009 | 4957 | 7 |

separately, which is collected in a different network environment than the training set. For other two datasets, we divide them into the training set and the test set according to the proportion of 80% and 20% for each class.

*4.1.2 Baselines.* We use seven state-of-the-art flow-level encrypted traffic classification methods covering three basic categories as baselines. For a fair comparison, all methods use the same partitioned flows for training and test.

- **AppScanner** (*Statistical features and ML-based model*).
  AppScanner [37] trains random forest classifiers by exploiting statistical features of packet sizes at flow-level. We retrained the ML-based model using the default hyper-parameter settings in their paper.
- **Decision Tree** (*Statistical features and ML-based model*).
  Gerard *et.al.* [12] trains C4.5 decision tree using statistical features of internal arrivals at flow-level. Likewise, we use the default settings.
- **Whisper** (*Statistical features and ML-based model*).
  Whisper [16] extracts the frequency domain features of packet sizes at flow-level as a supplementation of conventional statistical features, and uses clustering algorithms for classification. We reproduce Whisper on three datasets without modifications and then retrain the ML-based model.
- **Flowlens** (*Sequential attributes and ML-based model*).
  Flowlens [5] computes for each flow a memory-efficient representation of packet sizes named "flow marker", and uses a Multinomial Naive-Bayes classifier for classification. We retrained the ML-based model using the hyper-parameter settings that produce the most accurate results.
- **FS-Net** (*Sequential attributes and DL-based model*).
  FS-Net [26] uses recurrent neural networks (RNN) to automatically extract representations from raw packet size sequences. A multi-layer encoder-decoder structure and the reconstruction mechanism are adopted to enhance the effectiveness of features. We use the default hyper-parameter setting in their paper.
- **AttnLSTM** (*raw traffic and DL-based model*).
  AttnLSTM [42] is an end-to-end network based on the LSTM model to directly perform classification on raw traffic. It introduces an attention mechanism to score the importance of each flow. Similarly, we use the default setting in their paper.
- **ET-Bert** (*raw traffic and DL-based model*).
  ET-Bert [25] pre-trains deep traffic representations from large-scale unlabeled raw traffic, then fine-tunes on a small

**Table 2: Comparison Results on SJTU-AN21, ISCX-Tor, and ISCX-VPN datasets.**

| Dataset | SJTU-AN21 | | | | ISCX-Tor | | | | ISCX-VPN | | | |
|---|---|---|---|---|---|---|---|---|---|---|---|---|
| Method | AC | PR | RC | F1 | AC | PR | RC | F1 | AC | PR | RC | F1 |
| AppScanner [37] | 0.7181 | 0.7535 | 0.7181 | 0.7038 | 0.8203 | 0.8117 | 0.8203 | 0.8022 | 0.7293 | 0.7378 | 0.7293 | 0.7193 |
| Decision Tree [12] | 0.5702 | 0.6630 | 0.5702 | 0.5621 | 0.8059 | 0.7926 | 0.8059 | 0.7942 | 0.8259 | 0.8204 | 0.8259 | 0.8211 |
| Whisper [16] | 0.4820 | 0.5629 | 0.4820 | 0.5066 | 0.6723 | 0.7886 | 0.6723 | 0.6975 | 0.5848 | 0.6027 | 0.5848 | 0.5486 |
| Flowlens [5] | 0.6943 | 0.7576 | 0.6943 | 0.7128 | 0.8003 | 0.8703 | 0.8003 | 0.8256 | 0.6336 | 0.6674 | 0.6336 | 0.5820 |
| FS-Net [26] | 0.8083 | 0.8233 | 0.8083 | 0.7949 | 0.9322 | 0.9342 | 0.9322 | 0.9315 | 0.8457 | 0.8502 | 0.8457 | 0.8398 |
| AttnLSTM [42] | 0.8120 | 0.8176 | 0.8120 | 0.8030 | 0.9725 | 0.9718 | 0.9725 | 0.9708 | 0.9778 | 0.9781 | 0.9778 | 0.9778 |
| ET-Bert [25] | 0.8661 | 0.9163 | 0.8661 | 0.8815 | 0.9525 | 0.9514 | 0.9525 | 0.9445 | 0.9885 | 0.9895 | 0.9885 | 0.9888 |
| AN-Net (ours) | **0.9476** | **0.9490** | **0.9476** | **0.9439** | **0.9951** | **0.9951** | **0.9951** | **0.9950** | **0.9996** | **0.9996** | **0.9996** | **0.9996** |

amount of labeled data. We re-pretrain the model and fine-tune it on three datasets, respectively.

*4.1.3 Evaluation Metrics and Implementation Details.* We evaluate our AN-Net and compare it with other state-of-the-art methods by four typical metrics, including Accuracy (AC), Precision (PR), Recall (RC), and F1 [25, 38, 46]. In the training phase, we train the AN-Net with a stochastic gradient descent (SGD) optimizer, and the learning rate is set to 0.001. The batch size is 64 and the total steps is 50,000. The temperature hyper-parameter $\tau$ is set to 0.1. The probability of randomly drop uni-modal representation $p$ is set to 0.2 and the scaling factor $B$ is sampled from beta distribution $B \sim Be(4, 4)$. All the experiments are implemented using Pytorch 1.9.0 and trained on PC with Intel® Xeon® Gold 5218R CPU@2.10GHz, 256 GB RAM, and an NVIDIA GeForce RTX3090 GPU. The modal selection criteria is illustrated in Appendix B. For ISCX-VPN and ISCX-Tor datasets, we use Raw-SFEM for short-term feature extraction. For SJTU-AN21 dataset, we employ Stat-SFEM and drop the payload modality (see Section 4.5 for more details).

## 4.2 Comparison with State-of-the-Art Method

We compare AN-Net with seven state-of-the-art (SOTA) methods on three datasets. The experimental results are shown in Table 2. The seven methods can be devided into three categories: statistical feature-based methods (i.e., AppScanner, Decision Tree, and Whisper), sequential attribute-based methods (i.e., Flowlens and FS-Net), and raw traffic-based methods (i.e., AttnLSTM and ET-Bert).

**SJTU-AN21.** SJTU-AN21 [44] is a new anonymity network traffic dataset collected in the open network and the test set is collected separately in different network environment. Due to the interference of irrelevant packets and discrepancies between training and test data, previous methods never achieved more than 90% accuracy. According to Table 2, AN-Net significantly outperforms all existing methods. Specifically, AN-Net improves Accuracy and F1 by 8.15% and 6.24% respectively over the existing state of the art (i.e., ET-Bert). Previous methods designed for encrypted traffic classification ignored the noise of irrelevant packets and the unreliability of uni-modal attributes. As a comparison, we build strong short-term features and aggregate them with a carefully designed high temperature self-attention mechanism to resist irrelevant packet noise, and then propose to fuse representations from different modalities

to combat per-packet attribute noise. Moreover, Stat-SFEM exhibits strong transfer capabilities thanks to the high-level statistical features when the network environments of the training data and test data are inconsistent (see Section 4.5 for more details).

**ISCX-Tor.** ISCX-Tor [22] is a frequently used Tor network traffic dataset. Compared with SJTU-AN21, this dataset is less noisy and significantly larger in size. Because of the purity and large amount of data, DL-based methods that directly learn from raw sequential attributes or raw traffic payload (i.e., FS-Net, AttnLSTM, and ET-Bert) perform very well. AN-Net has an accuracy of 99.50% and a F1 of 99.51%, slightly better than these three DL-based methods, and significantly outperforms other four ML-based methods (i.e., AppScanner, Decision Tree, Whisper, and Flowlens). For example, AN-Net achieves 2.42% and 5.05% improvement on F1 over AttnL-STM and ET-Bert, respectively. Although the amount of data is large enough to support raw traffic-based methods to extract implicit and robust features from payload, attributes from other modalities can still help improve model performance.

**ISCX-VPN.** ISCX-VPN [12] is a commonly used VPN traffic dataset. Similar to anonymous networks, VPN networks use proxies to transmit information for hiding IP information. Therefore, VPN traffic classification also suffers from irrelevant packet noise. AN-Net pushes F1 on ISCX-VPN to 99.96%. Our model achieves more than 15.98% improvement on F1 over statistical-based methods and sequential attribute-based methods, and performs slightly better than two raw traffic-based methods (2.18% and 1.08% improvement on F1 over AttnLSTM and ET-Bert). Moreover, AN-Net exhibits greater robustness against injected noise packet attacks (see Section 4.3).

## 4.3 Robustness Analysis

To evaluate the robustness of AN-Net, we assume that attackers construct injected noise packet attacks, i.e., injecting irrelevant packets into original traffic to evade supervision. In the experiments, for simplicity, we assume attackers randomly select packet sequences from the entire dataset as noise traffic. We then mix original traffic with noise traffic in different ratios, i.e., the proportion of noise traffic ranges from 0 to 75%. We do not inject a higher proportion of noise traffic because the effectiveness of other methods is already low. Figure 4 shows F1 scores of AN-Net and seven SOTA methods

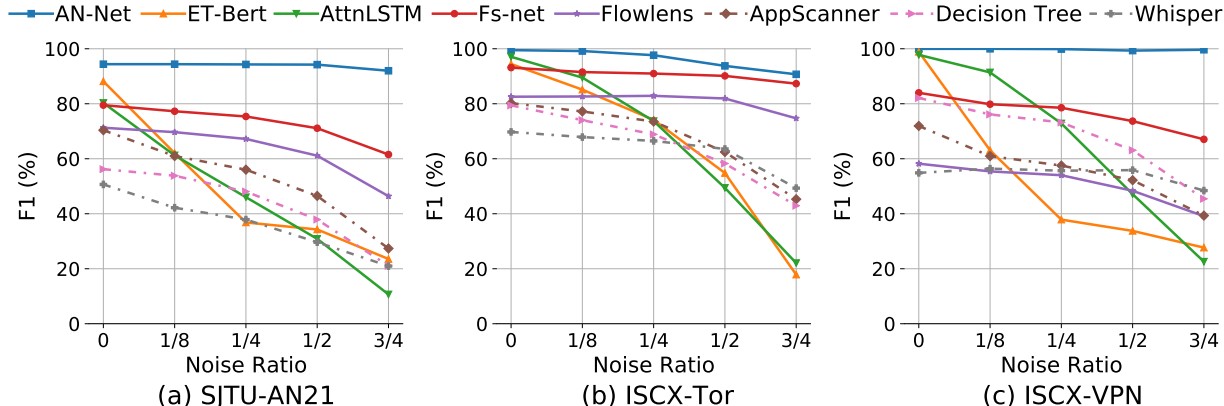

Figure 4: F1 score of AN-Net and seven SOTA methods under injected noise packet attacks.

Table 3: Ablation study on short-term features and high temperature (HT) self-attention mechanism.

| Feature | HT | Noise Ratio | | | |
|---|---|---|---|---|---|
| | | 0 | 1/4 | 1/2 | 3/4 |
| Long-Term | / | 0.9427 | 0.9276 | 0.9093 | 0.7295 |
| Short-Term | ✗ | 0.9352 | 0.9329 | 0.9327 | 0.9068 |
| | ✓ | **0.9439** | **0.9430** | **0.9423** | **0.9200** |

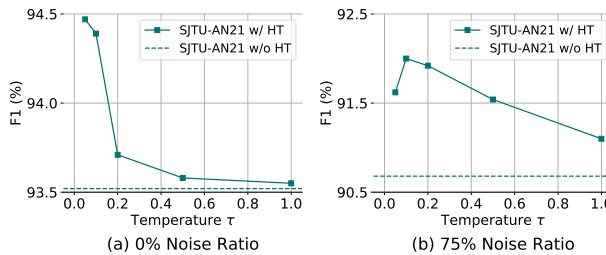

Figure 5: The impact of the temperature hyper-parameter $\tau$.

over three datasets under injected noise packet attacks. According to the results, we conclude that attackers cannot confuse AN-Net via injected noise packet attacks. However, attackers can fool other encrypted traffic classification models.

Raw traffic-based methods (i.e., AttnLSTM and ET-Bert) are very vulnerable to injected noise packet attacks. For instance, the F1 scores of ET-Bert and AttnLSTM on three datasets are reduced by at least 64.6% and 69.6% respectively. Payloads from irrelevant packets can easily lead to incorrect recognition results. Similarly, long-term statistical features (e.g., Maximum value, Mean value) can also be severely corrupted by inserted noise packets. Statistical feature-based methods (i.e., AppScanner, Decision Tree, and Whisper) have at most 43.0%, 36.5%, and 36.7% F1 score decrease over three datasets, respectively. As a comparison, sequential attribute-based methods (i.e., Flowlens and FS-Net) is more robust against irrelevant packet noise, since a part of clean original attribute sequences is retained. However, the F1 scores of FS-Net on the SJTU-AN21 and ISCX-VPN datasets still drop by 17.9% and 16.9%, respectively. In contrast, AN-Net maintains similar classification performance, where the F1 score fluctuations are less than 2.39% on SJTU-AN21 dataset and 0.32% on ISCX-VPN dataset. On ISCX-Tor dataset, due to the large amount of data, the F1 score of AN-Net drops by 8.8% to 90.71%, which is still higher than all other SOTA methods.

In summary, AN-Net can achieve robust classification because of the short-term representation learning and the multi-modal representation fusion. In particular, attackers cannot easily disrupt

all short-term features of all modalities and thus AN-Net is robust against injected noise packet attacks.

## 4.4 Ablation Analysis

We provide an ablation analysis to verify the contribution of each component on SJTU-AN21 dataset. In addition to normal traffic, we also perform ablation experiments under injected noise packet attacks to prove the effectiveness of these components against noise.

*4.4.1 Short-term Representation Learning.* In this section, we ablate short-term features and high temperature (HT) self-attention mechanism, as shown in Table 3. We do not perform flow division and directly extract long-term statistical features when ablating short-term features. Then we use vanilla self-attention mechanism to substitute high temperature self-attention mechanism. As discussed above (Section 4.3), long-term statistical features are severely corrupted by inserted noise packets. Results show that the F1 score of using long-term features is 19.05% lower than using short-term features when the noise ratio of irrelevant packets is set to 75%. Therefore, it is crucial to model short-term features to combat irrelevant packet noise. Besides, high temperature self-attention mechanism improves model performance through paying less attention to noise packets. This improvement increases as the noise ratio increases, from 0.87% to 1.32%. The visualization of high temperature self-attention mechanism is shown in Appendix 5.

We further investigate the impact of the temperature hyper-parameter $\tau$, as shown in Figure 5. Results show that AN-Net

**Table 4: Ablation study on multi-modalities.**

| Packet Size | IAT | TTL | TCPFlag | AC | F1 |
|:---:|:---:|:---:|:---:|:---:|:---:|
| ✓ | ✗ | ✗ | ✗ | 0.7958 | 0.8014 |
| ✓ | ✓ | ✗ | ✗ | 0.8314 | 0.8304 |
| ✓ | ✓ | ✓ | ✗ | 0.9269 | 0.9234 |
| ✓ | ✓ | ✓ | ✓ | **0.9476** | **0.9439** |

**Table 5: Ablation study on representation enhancement (RE) strategy.**

| RE | SJTU-AN21 | | ISCX-Tor | | ISCX-VPN | |
|:---:|:---:|:---:|:---:|:---:|:---:|:---:|
| | 1/2 | 3/4 | 1/2 | 3/4 | 1/2 | 3/4 |
| ✗ | 0.9378 | 0.9101 | 0.9334 | 0.8934 | 0.9957 | 0.9881 |
| ✓ | **0.9423** | **0.9200** | **0.9377** | **0.9071** | **0.9964** | **0.9933** |

achieves better performance at higher temperatures. Note that the real temperature is the reciprocal of temperature hyper-parameter $\tau$. The self-attention mechanism with a higher temperature generates a sharper distribution for weight score matrix, thus helps resist the irrelevant packet noise by paying little attention to noisy short-term features from irrelevant packets. When the temperature gradually decreases, the model performance gradually drops to the level of vanilla self-attention mechanism.

*4.4.2 Multi-modal Representation Fusion.* In this section, we first construct an ablation study on multi-modalities. As shown in Table 4, the uni-modal model using packet size only achieves an F1 score of 80.14%. By combining the attributes of other modalities, the F1 score of the multi-modal model is increased to 94.39% (14.25%↑). On the one hand, attributes of different modalities can provide complementary information to support better decision-making. On the other hand, when the network environment fluctuates, the attributes of different modalities can verify each other to combat per-packet attribute noise.

We further construct an ablation study on representation enhancement (RE) strategy. As shown in Table 5, representation enhancement strategy improves model performance through performing data augmentation in representation-level to combat the noise of uni-modal representations. The improvement also increases as the noise ratio increases, from 0.45% to 0.99% on SJTU-AN21 dataset and from 0.43% to 1.37% on ISCX-Tor dataset. The unreliability of per-packet attributes and the noise of irrelevant packets both make the uni-modal representation full of noise. The representation-level data augmentation strategy enables AN-Net to learn robust multi-modal representation from noisy uni-modal representations.

## 4.5 Stability Analysis

In this section, we provide an stability analysis to demonstrate that statistical features are more stable than raw attribute sequences or raw traffic payloads when training data collection environment is inconsistent with the actual test environment.

**Table 6: Comparison of Stat-SFEM and Raw-SFEM and ablation of payload modality on SJTU-AN21 dataset.**

| SFEM | Payload | AC | F1 |
|:---:|:---:|:---:|:---:|
| Raw-SFEM | ✗ | 0.9055 | 0.9098 |
| Stat-SFEM | ✓ | 0.9358 | 0.9391 |
| | ✗ | **0.9476** | **0.9439** |

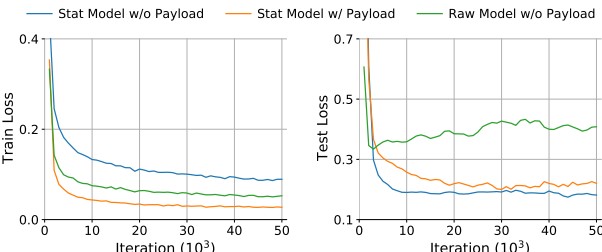

**Figure 6: Training loss and test loss curves during training.**

Specifically, we construct an comparison experiment to compare Stat-SFEM and Raw SFEM, and then perform an ablation experiment on the payload modality over SJTU-AN21 dataset. Results are shown in Table 6. It can be concluded that Stat-SFEM outperforms Raw-SFEM by a large margin (4.21% in Accuracy and 3.41% in F1 score) and adding payload modality slightly reduces model performance (1.18% in Accuracy and 0.48% in F1 score). We further plot the training loss and test loss curves during the training process, as shown in Figure 6. Stat model has the largest training loss, but the smallest test loss, which indicates that learning from raw attribute sequences or raw traffic payloads will suffer from severely overfitting. When training data collection environment is inconsistent with the actual test environment, some specific raw attribute sequences or raw traffic payloads that are very useful in training data may become ineffective in test data. For example, changes in encryption algorithms result in different payloads for the same plaintext. In contrast, high-level statistical features are more stable because they measure the distribution and variation of raw attribute sequences, and thus are more transferable.

## 5 CONCLUSION

In this paper, we propose a new anonymous traffic classification model, AN-Net, to construct robust short-term representations for a single modality and then combine representations from different modalities. AN-Net is able to resist irrelevant packet noise and per-packet attribute noise, thus exhibits strong robustness against injected noise packet attacks. We comprehensively evaluate the effectiveness and generalization of AN-Net on two anonymous traffic datasets and one VPN traffic dataset. Experimental results show that AN-Net achieves a new state-of-the-art performance, notably eleveting the F1 score of SJTU-AN21 to 94.39% (6.24%↑). Moreover, AN-Net is more robust than existing works against injected noise packet attacks, because attackers cannot easily disrupt all short-term features of all modalities.

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

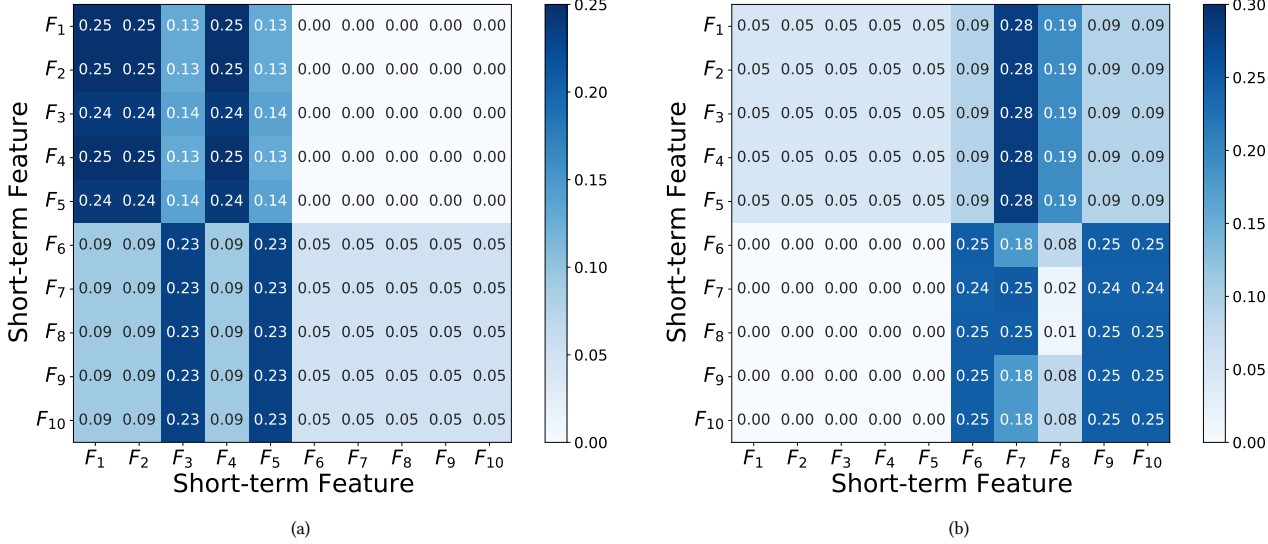

(a)

(b)

Figure 7: The visualization of the weight score matrices in the high temperature self-attention mechanism. Results show that the high temperature self-attention mechanism generates a sharp distribution for weight score matrix and all short-term features (including noisy short-term features) pay little attention to noisy short-term features from irrelevant packets.

## A VISUALIZATION OF HT SELF-ATTENTION.

We visualize the weight score matrix to exhibit how high temperature self-attention mechanism helps resist irrelevant packets noise. Specifically, we first construct a noisy dataset containing 50% irrelevant packets by injecting irrelevant packets, and then train AN-Net with this noisy dataset. The weight score matrices of two special text examples are shown in Figure 7. Each flow is divided into 10 short-term packet sequences. In Figure 7(a), the left half is the target packet sequences, and the right half is the irrelevant packet sequences, which is exactly the opposite of Figure 7(b).

It can be concluded that the high temperature self-attention mechanism generates a sharp distribution for weight score matrix and all short-term features (including noisy short-term features) pay little attention to noisy short-term features from irrelevant packets. The high temperature self-attention mechanism increases the magnitude of dot products by the temperature hyper-parameter $\tau$. After employing a high temperature, the softmax function can produce extremely small weights for noisy short-term features from irrelevant packets (0.00 or 0.05 in Figure 7). By combining short-term features with high temperature self-attention mechanism, AN-Net is effective to resist irrelevant packet noise.

## B MODAL SELECTION.

As mentioned above, before fusing representations from different modalities, we resort to information leakage to remove useless modalities. We compute the mutual information between statistical features of a certain modalities and the ground truth labels to measure the importance of this modality. The calculation results and selection strategies are shown in Table 7. Modalities with lower information leakage are removed to reduce model complexity. Specifically, we remove the IPFlag modality and TTL modality

Table 7: Mutual information and selection strategies of each modality on three datasets.

| Dataset | Metric | Packet Size | IAT | TTL | IPFlag | TCPFlag |
|---------|--------|-------------|-----|-----|--------|---------|
| SJTU-AN21 | MI | 0.81 | 0.81 | 1.50 | 0.01 | 0.99 |
| | Selection | ✔ | ✔ | ✔ | ✗ | ✔ |
| ISCX-Tor | MI | 1.14 | 0.82 | 0.00 | 0.95 | 0.84 |
| | Selection | ✔ | ✔ | ✗ | ✔ | ✔ |
| ISCX-VPN | MI | 0.68 | 0.75 | 1.34 | 0.23 | 0.42 |
| | Selection | ✔ | ✔ | ✔ | ✗ | ✗ |

for SJTU-AN21 and ISCX-Tor datasets, respectively. For ISCX-VPN dataset, we drop the IPFlag and TCPFlag modalities.

