# OpenReview forum: "AN-Net: An Anti-Noise Network For Anonymous Traffic Classification"
_ACM.org/TheWebConf/2024/Conference — TheWebConf24_

### Official Review · Reviewer_HFsD · 2023-10-28

**Novelty:** 4
**Technical Quality:** 4

**Review:**

This submission tries to study an anti-noise network for traffic classification with an enhanced multi-modal fusion approach

pros
1. this draft is easy to follow
2. your core idea of uni-modal short-term representation is reasonable
3. your SRAM is similar and closely linked with clustering

cons
1. your three data sets are too small
2. missing baselines, some state-of-the-art online or real-time traffic classification methods are not tested
3. what's the practical usefulness of this proposed model is unclear

Overall, it's enjoyable to have a read on this draft, it would be glad to recommend towards acceptance, after this work thoroughly be polished based on all comments.

**Questions:**

1. your data source is too few, try to add at least one or two
2. the number of data points in your each experiment is too small, try to significantly enlarge
3. try to introduce some production data from companies, etc, to better demonstrate the usability of the proposed model
4. there are related state-of-the-art you may want to compare: Online Optimization Methods for the Quantification Problem, Improved Algorithm on Online Clustering of Bandits
5. you are strongly encouraged to add some theoretical analysis and guarantee to make this work more solid

**Reviewer Confidence:**

4: The reviewer is certain that the evaluation is correct and very familiar with the relevant literature

**Scope:**

4: The work is relevant to the Web and to the track, and is of broad interest to the community

---

### Official Review · Reviewer_Nmxv · 2023-11-09

**Novelty:** 6
**Technical Quality:** 6

**Review:**

This paper presents an anonymous traffic classification model. It constructs robust short-term representations for a single modality and combines representations from different modalities for flow representation learning. The proposed method can resist irrelevant packet noise and per-packet attribute noise. Extensive experiments are conducted to evaluate the effectiveness of the proposed methods. The experimental results verify the effectiveness of different components of the proposed model.

Strength:
1) The paper is well written and easy to follow.
2) The assumption is reasonable, and the experimental results are convincing.
3) The experiments are sufficient and solid.

Weakness:
1) Some symbols are repeatedly used, e.g., $C$ in Equation (9).
2) In Section 4.3, how the original traffic is mixed with noise traffic should be clarified.
3) The objective function for model training is missing and more details about the training process are desired.

**Questions:**

Can the method correctly classify traffic with short flows?

**Ethics Review Description:**

No ethical issues

**Reviewer Confidence:**

3: The reviewer is confident but not certain that the evaluation is correct

**Scope:**

3: The work is somewhat relevant to the Web and to the track, and is of narrow interest to a sub-community

---

### Official Review · Reviewer_7HCC · 2023-11-23

**Novelty:** 4
**Technical Quality:** 5

**Review:**

This paper proposes a method for network traffic classification.  Existing methods are not robust against noise, and the idea behind this work is to use a short-term representation so that irrelevant packet noise has less of an effect.  Experiments on real datasets show good results.  On the whole, I thought that this paper was well-written, easy to follow, and the experimental analysis was mostly thorough (though see comments/questions below).  I appreciated the ablation study, and many baselines were included.

My biggest concern is about the nature of the method's robustness.  I understand that the goal of the method is to be robust against injected noise, so it makes sense that the experimental evaluation is concentrated about that.  Moreover, the paper justified that this is a legitimate complication for the classification task.  But are there other challenges associated with accurate classification, and how well does your method perform in the face of such challenges?  Even if it is designed specifically for injected noise, it is important to know how it compares to other methods with respect to robustness in other ways, or even with respect to robustness against different types of injected noise.

Thanks for the response.  It clarified some of my concerns.

**Questions:**

See my question about method robustness above.

**Ethics Review Description:**

-

**Reviewer Confidence:**

2: The reviewer is willing to defend the evaluation, but it is likely that the reviewer did not understand parts of the paper

**Scope:**

3: The work is somewhat relevant to the Web and to the track, and is of narrow interest to a sub-community

---

### Official Review · Reviewer_M9Dg · 2023-11-23

**Novelty:** 5
**Technical Quality:** 4

**Review:**

This paper proposes an anonymous traffic classification model, AN-Net. The model constructs robust short-term representations and then combines representations from different modalities. AN-Net is able to resist irrelevant packet noise and per-packet attribute noise. The authors evaluated AN-Net on two anonymous traffic datasets and one VPN traffic dataset, and AN-Net achieves state-of-the-art performance. In addition, the author conducted extensive analysis to demonstrate the robustness of the proposed model against attacks and analyzed the impact of different data on performance. Below are my main comments.

S1: An effective model that achieved SOTA performance in the experiment.

S2: Provided multi-dimension analysis for the proposed model, i.e., the contribution of each component and the robustness and stability of the AN-Net.

W1: For those who are not familiar with traffic classification, understanding this paper is very difficult. The motivation behind the classification of anonymous traffic is not well explained. For example, what obstacles do encryption mechanisms present to traffic classification, and why do they aid malicious traffic in eluding the surveillance system? Which challenges did this paper address? (Line 91-93).

W2: The rationale behind the model design can be further elucidated. In particular, the author combats per-pack attribute noise through multi-modal representation fusion but does not provide a reason. There are many ways to represent fusion (e.g., concatenation). Why is the proposed method more conducive to dealing with per-packet attribute noise? Readers hope to gain more insights into model design.

W3: Some symbols lack explanation. For example, the meaning of the superscript C is unclear. The superscript D also does not give a description in detail.

**Questions:**

See above weakness.

**Reviewer Confidence:**

3: The reviewer is confident but not certain that the evaluation is correct

**Scope:**

3: The work is somewhat relevant to the Web and to the track, and is of narrow interest to a sub-community

---

### Official Review · Reviewer_DHEL · 2023-11-24

**Novelty:** 6
**Technical Quality:** 6

**Review:**

### Summary
- **Anonymous traffic classification**: The author aims to classify network traffic from various applications or web services in anonymous networks, such as Tor or VPN, that use proxies to protect user privacy and make traffic packets from different sources form a single flow.
- **Noise resistance**: The paper identifies two types of noise that hinder the classification performance: irrelevant packet noise, which refers to packets from other flows that are mixed with the target flow, and per-packet attribute noise, which refers to fluctuations in the packet attributes (such as size, arrival time, etc.) due to network conditions.
- **AN-Net**: The author proposes an Anti-Noise Network (AN-Net) to extract robust patterns from noisy traffic data. AN-Net consists of two main modules: a Uni-modal Short-term Representation Learning Module, which divides a flow into short-term packet sequences and aggregates them with a novel high-temperature self-attention mechanism, and an Enhanced Multi-modal Representation Fusion Module, which fuses representations from different packet attributes and performs representation enhancement to improve the robustness.
- **Experiments and results**: The paper evaluates AN-Net on two anonymous traffic datasets and one VPN traffic dataset and compares it with seven state-of-the-art methods. AN-Net achieves the best performance on all datasets and shows strong robustness against injected noise packet attacks. The paper also conducts ablation studies and analyses to demonstrate the effectiveness of the critical components of AN-Net.

### Strengths
1. The paper addresses a novel and challenging problem of anonymous traffic classification, which has not been well-studied in the literature. The author proposes a novel and effective method that can handle the noise issues in anonymous traffic and achieves state-of-the-art performance and robustness on three datasets.
2. The paper conducts extensive experiments on three datasets and demonstrates the superiority and generalization of the proposed method over existing methods. It also shows the effectiveness of the proposed method against injected noise packet attacks and the stability of the statistical features in different network environments.
3. The paper is well-written and clear.

**Questions:**

1. Could you provide a theoretical analysis or justification for the proposed method, such as why the high temperature self-attention mechanism and the representation enhancement strategy can improve the performance and robustness of the model?
2.  Have you compared the proposed method with other state-of-the-art methods for encrypted traffic classification, which may have different characteristics and challenges than anonymous traffic classification?
3. Could you discuss the limitations and risks of the proposed method, such as the possible countermeasures or attacks that can evade or degrade the performance of the proposed method?
4. How does the proposed method handle the dynamic changes in network conditions, such as changes in bandwidth, latency, or packet loss rate?
5. How does the proposed method deal with the diversity and heterogeneity of anonymous traffic, such as different types of applications, protocols, or user behaviors?
6. Could you provide more details about the implementation and computational complexity of the proposed method, such as the training time, inference time, memory usage, or scalability?
7. Have you considered the privacy implications of the proposed method, such as whether it could inadvertently reveal sensitive information about the users or violate their privacy rights?
8. How does the proposed method handle the evolution and advancement of anonymous networks, such as new versions or features of Tor or VPN that could change the characteristics or patterns of anonymous traffic?
9. Have you considered extending the proposed method to other types of network traffic or other types of noise, such as encrypted traffic, mobile traffic, or measurement noise?

**Reviewer Confidence:**

1: The reviewer's evaluation is an educated guess

**Scope:**

3: The work is somewhat relevant to the Web and to the track, and is of narrow interest to a sub-community

---

### Decision · Program_Chairs · 2024-01-22

**Decision:**

Accept

**Comment:**

The paper proposes an approach to classify anonymous traffic. The main contributions over SOTA are a robust short-term representation learning step, and fusing of information from different modalities, which grants the approach its robustness to irrelevant packet noise and per-packet attribute noise. Extensive experiments considering multiple datasets and baselines, as well as ablation studies demonstrate its empirical utility.

 Most of the scores of the reviewers are border or slightly above acceptance threshold. The major review concerns that stand out are: (1) robustness of proposed approach to noises and attacks that may be present in real-world production data that are not captured in the benchmark and hence not incorporated in the design of the approach, (2) missing comparison to SOTA pointed out by Reviewer HFsD. Therefore, I recommend a borderline/weak accept.